# Occurrence of Free-Living Amoebae in Non-Human Primate Gut

**DOI:** 10.3390/tropicalmed9050108

**Published:** 2024-05-08

**Authors:** Igor Rodrigues Cardoso, Clezia Siqueira de Lima, Rhagner Bonono dos Reis, Ana Cristina Araujo Pinto, Thalita Pissinatti, Tatiana Kugelmeier, Sócrates Fraga da Costa Neto, Fabio Alves da Silva, Helena Lúcia Carneiro Santos

**Affiliations:** 1Laboratório de Doenças Parasitárias, Instituto Oswaldo Cruz/FIOCRUZ, Rio de Janeiro 21040-360, Brazil; igor.car.doso@hotmail.com (I.R.C.); cleziasiqueira@hotmail.com (C.S.d.L.); bonono.rhagner@ioc.fiocruz.br (R.B.d.R.); 2Instituto de Saúde de Nova Friburgo, da Universidade Federal Fluminense, Nova Friburgo 28625-650, Brazil; 3Instituto de Ciência e Tecnologia em Biomodelos/FIOCRUZ, Rio de Janeiro 26382-462, Brazil; ana.cristina@fiocruz.br (A.C.A.P.); thalita.pissinatti@fiocruz.br (T.P.); tkugelmeier@gmail.com (T.K.); fabiosilva@fiocruz.br (F.A.d.S.); 4Fiocruz Mata Atlântica, Fundação Oswaldo Cruz, Rio de Janeiro 22713-560, Brazil; socratesfneto@yahoo.com.br

**Keywords:** microbiome, culture, morphology, *Acanthamoeba*, *Vermamoeba*

## Abstract

The gut microbiome reflects health and predicts possible disease in hosts. A holistic view of this community is needed, focusing on identifying species and dissecting how species interact with their host and each other, regardless of whether their presence is beneficial, inconsequential, or detrimental. The distribution of gut-associated eukaryotes within and across non-human primates is likely driven by host behavior and ecology. To ascertain the existence of free-living amoebae (FLA) in the gut of wild and captive non-human primates, 101 stool samples were collected and submitted to culture-dependent microscopy examination and DNA sequencing. Free-living amoebae were detected in 45.4% (46/101) of fecal samples analyzed, and their morphological characteristics matched those of *Acanthamoeba* spp., *Vermamoeba* spp., heterolobosean amoeboflagellates and fan-shaped amoebae of the family Vannellidae. Sequence analysis of the PCR products revealed that the suspected amoebae are highly homologous (99% identity and 100% query coverage) with *Acanthamoeba* T4 genotype and *Vermamoeba vermiformis* amoebae. The results showed a great diversity of amoebae in the non-human primate’s microbiome, which may pose a potential risk to the health of NHPs. To our knowledge, this is the first report of free-living amoebae in non-human primates that are naturally infected. However, it is unknown whether gut-borne amoebae exploit a viable ecological niche or are simply transient residents in the gut.

## 1. Introduction

The complex network of microorganisms in the intestinal microbiome regulates host health through immune, endocrine, and metabolic pathways (e.g., food decomposition, nutrient intake, and drug metabolism) [1,2]. Emerging evidence shows that most of the members of this complex remain unknown [3,4]. New insights into the composition of this community are crucial for determining its potentially significant implications for animal health and biological diversity. Despite recent advances in eukaryotic enrichment protocols of metagenomics approaches, studies focused on microeukaryotic organisms are still a young field [5,6]. Indeed, the databases, analysis of the data, and assembly-based metagenomic tools have needed to be more robust when targeting eukaryotic microbes [7]. Lately, a study reported that captive non-human primates (NHPs) harbor microbial species that are more similar to human ones compared to wild NHPs, and the extent of microbiome overlap is firmly lifestyle-dependent [8]. Several parasites, such as *Babesia*, *Cryptosporidium*, amoebae, *Toxoplasma*, *Trypanosoma*, coccidia, nematodes, and cestodes, are found in the NHP gut possibly constituting a risk for primates, including humans [9,10,11]. The gut parasite *Entamoeba* does have a profound effect on humans (*Entamoeba histolytica* kills > 55,000 people each year) and other animals. Despite it belonging to the Amoebozoa (like *Vermamoeba* and *Acanthamoeba*) it is not a free-living amoeba (FLA) but is rather an obligate parasite.

Recently, the DNA of free-living amoebae (FLA) has been detected by classical approaches and PCR amplification, along with high-throughput sequencing in fecal samples [12]. However, only some studies have attempted to address the presence of FLA in the gut microbiome of mammals and other animals [13,14,15,16,17]. FLAs are aerobic, mitochondrial, underexplored eukaryotic protists ubiquitous in diverse environmental sources [18,19,20]. Among protists, free-living amoebae are the dominant bacterial consumers [21]. The richness of bacteria in the gut may favor natural colonization or a transitory passage in the intestinal microbiome. Recently, a metagenomic analysis revealed the presence of the Acanthamoebidae and Vahlkampfiidae families in the microbiome of NHPs, and other studies have reported the presence of FLAs in the intestine [12]. Emerging evidence has linked several FLAs to human disease. Some species of the genera *Acanthamoeba* spp., *Balamuthia* spp., *Sappinia* spp., and *Naegleria* spp. are potentially pathogenic for humans and other animals [22,23,24,25,26,27,28,29]. *Acanthamoeba* spp. are found in an aquatic environment, soil, and air and can be isolated from the rhinopharynx and the stool of healthy humans [13,14,15,19,22,23,24,25]. In recent years, the frequency of *Acanthamoeba* keratitis has increased worldwide after extended wear of soft contact lenses. On the other hand, brain or disseminated infections caused by *Acanthamoeba* sp. in immunocompromised individuals are considered rare [28,29]. Primary amoebic meningoencephalitis (PAM), caused by *Naegleria fowleri* and granulomatous amoebic encephalitis (GAE) caused by *Balamuthia mandrillaris*, are relatively rare too, but perhaps they are under-detected and thus, underestimated [18,29]. Indeed, clinical diseases have a poor prognosis. The symptoms are non-specific, which can be mistaken for other bacterial and viral diseases, and the clinicians who treat the patients are often not familiar with FLA [29]. And not surprisingly, almost all cases were diagnosed at autopsy. Additionally, reliable diagnostic tests are unavailable, and diagnostic expertise is limited [18,28,29].

Despite the issues highlighted above, FLAs have gained public attention in recent years due to their capability to harbor pathogenic and nonpathogenic microorganisms through transient or symbiotic relationships [30,31,32,33]. Moreover, due to their resistance to chlorine, the amoebic cysts are considered vectors of viruses, bacteria, fungi, and protozoa [34]. Thus, FLAs have the potential to act as vectors for the transmission of several microorganisms. However, *Acanthamoeba* sp. interacts with many microorganisms [32,33,35]. *Legionella pneumophila* can infect, replicate, and kill the *Acanthamoeba* sp. [36]. While most studies investigate the interactions of *Acanthamoeba* with several microorganisms, other free-living amoebae are not assessed. Recently, *Vermamoeba vermiformis* has shown the potential to not only carry and allow *Helicobacter pylori* multiplication, but also to revive the bacteria to a culturable state [37]. It is essential to mention that the presence of FLAs in the intestinal microbiome has been reported but requires further investigation. It is becoming increasingly evident that the gut microbial community is a niche where *Acanthamoeba* and pathogens may interact, and further research should aim to better understand the interactions between FLAs and gut microbiota. The recent advancement in “culture-independent” methods of microbial characterization and high-throughput sequencing in fecal samples have proven highly valuable, although they do not directly address the most critical questions concerning host–microbiota interactions. To comprehensively understand the presence of FLAs in animals’ gut microbiota, the establishment of stable cultures is necessary to elucidate microbiota ecology in relation to the biology, potential pathogenicity, and FLA–microbiota interactions. This study employed culture-dependent approaches to isolate FLA from the gut microbiome, combined with morphological analysis, different staining techniques, and DNA sequencing. This study highlights the importance of FLA in the gut of mammals, with potential relevance to future studies on mucosal immunology and the potential pathogenicity of FLA. Additionally, human-pathogenic, amoeba-resisting microorganisms can be considered an important emerging field of study.

## 2. Materials and Methods

### 2.1. Sampling

Fecal samples were collected under three conditions: (i) samples obtained after defecation on the floor surface of cages; (ii) rectal washouts of captive non-human primates after attaining deep anesthesia, and (iii) samples directly collected from the intestine of wild animals after euthanasia.

Samples in this cohort were collected from 101 individuals representing three primate species of captive non-human primates. A total of 96 fecal samples were collected, including *Macaca mulata* (Rhesus macaque, N = 65), *Saimiri ustus* (Bare-eared squirrel monkey, N = 01), and *Saimiri sciureus* (South American Squirrel monkey, N = 10). Additionally, 20 rectal washout samples were obtained from *M. mulatta* at the Institute of Biomodel Science and Technology (ICTB, Fiocruz). Feces samples from captive animals were collected as soon as possible after defecation on the floor surface of their cages to minimize contamination, or directly from the rectum under anesthesia. In addition to these, five stool samples from *Callithris* sp. (N = 5) were obtained from the Fiocruz Atlantic Forest Campus of the Oswaldo Cruz Foundation (CFMA) (22°56′18″ S 43°24′11″ W). These stools were collected in the field laboratory after dissection of the viscera, stored in labeled plastic containers, and transported to the main laboratory. All samples were collected from animals habituated to humans, making it possible to collect them from specific individuals.

### 2.2. Culture and Morphological Analyses

Briefly, 1 g of fecal material was homogenized in 2 mL of phosphate-buffered saline; 50 µL of the homogenate, filtered through membrane filters with an 8 µm pore size, were added to 1.5% non-nutrient agar (NNA) plates containing 2 mL of Page’s amoeba saline without killed *Escherichia coli*. All plates were sealed and incubated at 27 °C. After two days, the plate was washed three times with media Page’s amoeba saline to remove the organic debris and bacteria layer. Subsequently, media Page’s and heat-killed *E. coli* were added to the plate. Every two days, the plate was examined using an inverted microscope. When positive, it was sub-cultured by cutting out a small piece of agar and placing it onto a fresh plate overlaid with a layer of heat-killed *E. coli*. Finally, within a week to two weeks, a robust culture was obtained, which can be maintained by periodic transfer of aliquots to new cultures. The plates were observed daily for amoebic growth up to 30 days after inoculation using an inverted microscope at 200× and 400× magnification.

The presence of FLAs was confirmed through the cyst and trophozoite morphology, and positive cultures were sub-cultured by cutting out a small piece of agar and placing it onto a fresh plate overlaid with a layer of heat-killed *E. coli*. These positive cultures were then used to study the morphology and motion characteristics of each isolate using both bright-field and phase-contrast microscopy, along with permanent stained smears. The material from the surface of the plate was removed, fixed, and stained with Giemsa [38], and Panoptic^®^ staining, then examined at a higher magnification (1000×). The smears were immersed for five seconds in each solution: A (fixative number 1), B (Eosin Panoptic No. 2), and C (Blue Panoptic No. 3) without any washes in between. Excess reagent was drained from the slides between solutions and the buffer solution, pH 7.2, for one second [39]. Additionally, a cloning methodology was employed wherein only a single cell was seeded in a new culture medium and the culture pellet was preserved at −20° C.

### 2.3. Molecular Identification of Free-Living Amoeba DNA Sequencing

Samples with a positive result for FLAs at microscopy were reassessed using a direct PCR method. DNA extraction from the isolates was performed with the commercial QIAmp DNA Mini *Kit* (Qiagen, Hilden, Germany), according to the manufacturer’s instructions. PCR assay was performed according to the morphological criteria of the amoebae in positive plates using different sets of primers. The genus *Acanthamoeba* was confirmed through a polymerase chain reaction (PCR), using forward primer JDP1 (5’-GGCCCAGATCGTTTACCGTGAA-3′) and the reverse primer JDP2 (5′-TCTCACAAGCTGCTAG GGAGTCA-3′) which amplifies a 423 to 551 bp fragment for the 18 S rDNA [40]. Likewise, PCR was performed for *Vermamoeba* spp. and *Vannella* spp. using the primers: forward NA1 5-GCTCCAATAG CGTATATTAA-3 and reverse NA2 5-AGAAAGAGCTATCAATCTGT-3 [41]. Fragments of approximately 650 and 700–800 bp were identified in gel electrophoresis. Amplicons were purified using the Wizard^®^ SV gel and PCR Clean-Up System kit (Promega, Madison, WI, USA) and sequenced for both strands using the PCR primers. DNA cycle sequencing reactions were performed using the BigDye^®^ Terminator v.3.1 Cycle Sequencing Kit and loaded in the ABI 3730 Sequencing Platform (both—Applied Biosystems, Foster City, CA, USA). Raw bidirectional sequences reads were trimmed, assembled into contigs, and manually edited using SeqMan (DNASTAR software package, DNASTAR Inc., Madison, WI, USA), and then exported in FASTA format. The consensus sequences were compared with previously published sequences using the Basic Local Alignment Search Tool (BLASTn) available in the GenBank sequence database.

### 2.4. Assessment of Viability

*Acanthamoeba castellanii (ATCC* NEF 30010) and *Acanthamoeba polyphaga (*ATCC 30461) trophozoite growth were cultured at 37 °C in the peptone-yeast extract-glucose medium (PYG). To assess the impact of acidic exposure on amoebae viability, trophozoites were incubated in media of over a pH range (pH of 2.0 to 6.0). The pH levels were adjusted to the required values (2, 3, 4, 5, and 6) and regulated automatically by HCl. Cultures were incubated for 12, 24, 36, 48, 72, and 96 h at 37 °C, with a starting density of 5 × 10^4^ amoebae/mL. Trophozoite density was determined by cell counting under a light microscope using a Neubauer chamber hemocytometer.

## 3. Results

Out of 101 fecal samples obtained from PNHs, 46 (45.4%) were positive for FLAs based on the morphological and locomotion criteria, as observed through inverted microscopy (magnification ×200 and ×400) using standard taxonomic identification sources. A significant diversity of amoebae was observed in the fecal samples of *Callithrix* sp. (N = 5), *Macaca mulatta* (N = 26), and *Saimiri sciureus* (N = 8), as well as in rectal washout samples of *Macaca mulatta* (N = 7). All samples had mix-species infections where two or more morphotypes were associated in the same culture. Unfortunately, several cultures of FLA were lost due to significant fungal overgrowth, even with anti-fungal drugs.

Altogether, different groups of FLA were identified, including *Acanthamoeba* sp., *Vermamoeba* sp, heterolobosean amoeboflagellates, and fan-shaped amoebae of the *family Vannellidae* (Figure 1B–E). The trophozoites of the heterolobosean amoeboflagellates assumed a monopodial form, and the amoeboid-form organism changed to the transient, flagellate form with flagella at the broad end (Figure 1D). In addition, typical mobility was observed with either spinning or jerky movements (Appendix A), and a cyst form with thick single-wall structure with an outer gelatinous layer was observed.

*Acanthamoeba* sp. were identified in the form of double-walled cysts. The ectocyst (outer wall) was differentiated from the variably stained surrounding background, and the endocyst (inner wall) with a stellated, polygonal, square, round, or oval aspect was visually distinguished from the spherical outline of the ectocyst (Figure 2I–K). Unidentified amoeba with rounded cysts (Figure 2L) and mixed-species infections where two or more morphotypes were associated in the same culture, were also observed.

Numerous spine-like pseudopods (acanthopodia) gave the cell a spiny appearance (Figure 1E, Figure 3A–D,P,Q), with a single nucleus with a well-defined central nucleolus visible in the trophozoites (Figure 3B). Flotation forms of amoebae (Figure 3E,F) and fan-shaped amoebae of the *family Vannellidae* (Figure 3G–I) were observed. Unidentified trophozoites were also present.

All sample positives were mixed with their different genera. Subculturing facilitated isolation however, despite our efforts, only five samples were successfully mono-isolated.

The results of permanent-stained slides showed that the most consistent stain for identifying *Acanthamoeba* cysts and trophozoites was the Panoptic^®^ stain, followed by the Giemsa stain. The Giemsa stain gave poor visibility for acanthopodia. Trophozoite, flagellate, and cyst forms of heterolobosean amoeboflagellates were observed. The *Vermamoeba*-like trophozoite was detected in 2 of the 46 positive samples, (Figure 1C), displaying a slightly oval appearance with a single wall cyst, and the trophozoite had well visible cylindrical monopodia in the medium. In turn, fan-shaped amoebae of the *family Vannellidae* were identified by locomotive and floating forms, which are the main characteristics for this amoebae genus (Figure 3E–I). Moreover, trophozoites exhibited a semi-circular or fan-shaped morphology, with pronounced areas of the frontal hyaloplasm and the cytoplasm filled with numerous granules (Appendix A), containing numerous optically empty vacuoles (Figure 1B). Out of the total 46 samples positive for FLAs from captive and free-living PNHs cultivated in non-nutrient agar, only 36 were submitted to DNA amplification by PCR. The rest of the samples (N = 10) were not tested due to excessive fungal growth. PCR using genus-specific primers (JDP1 and JDP2) confirmed *Acanthamoeba* spp. in 77.8% (36/28), amplifying an expected fragment of approximately 500 bp. Sequencing of the PCR products and BLAST analysis revealed that 18% (5/28) of sequences belonged to the T4 genotypes when compared to the reference sequences deposited at GenBank. The percentage of identity ranged from 99% to 100% (accession number: MK713911, MT378247, MT378246, MF197422, MF100900, MT378235). The rest of the sequencing of the PCR products showed chromatograms suggestive of the presence of mixed infections. On the other hand, when the set of primers NA1/NA2 were used, 800 bp PCR products were obtained in 92% (33/36) isolates. However, mixed infections were clearly detected visually in the chromatograms with the sequencing trace showing two or more peaks in the same location. Out of the 33 PCR products, only one product was successfully sequenced. The sequence analysis revealed that the amoeba had a high homology of 99% to *V. vermiformis* (access number: KX856374.1, KP792393.1, MN238712.1). Moreover, the PCR assay failed to show any positive results for heterolobosean amoeboflagellates.

Across the entire pH range studied, remarkably, at an acidic pH of 2, *A. castellanii* and *A. polyphaga* were observed moving for 24 h but became nonviable within 48 h. However, at pH levels of 3 through 6, the amoebae remained viable for more than 96 h.

## 4. Discussion

Free-living amoebae are microbial eukaryotes that are widely distributed in the natural environment. In the last decade, these amoebae have attracted considerable research interest, mainly as environmental hosts of several intracellular pathogens [42]. However, our understanding of the role of free-living amoebae FLAs in the complex and extreme habitat of the mammal’s gut is still in its early stages. Arguably, FLAs are an underappreciated group within microbiota, and their interaction with the immune system is an open question, as well as the risk that this could pose to mammals. However, there have been no studies specifically focused on this issue. Recently, a report based on metagenomic analysis described the presence of Acanthamoebidae and Vahlkampfiidae families and other protists in the microbiome of NHPs [12]. Conversely, additional studies based on stool cultures have confirmed the presence of FLA in the guts of other vertebrates and invertebrates [13,14,15,16].

In the present study, an unexpected diversity of free-living amoebae in the fecal microbiota of non-human primates was observed. Among the many free-living amoebae existing in nature, *Acanthamoeba* spp., *Balamuthia mandrilaris*, *Vermamoeba vermiformis*, *Naegleria fowleri*, and *Sappinea pedata* have been much more studied due to their association with diseases [19,21,43]. A case of fulminant amoebic meningoencephalitis and pneumonitis in a simian immunodeficiency virus (SIV)-infected rhesus macaque was reported due to *Acanthamoeba* spp. [44]. In turn, *B. mandrillaris* was first discovered in a mandrill baboon (*Papio sphinx*) that died of encephalitis at the San Diego Zoo Wildlife Park in California in 1986 [45]. In further studies, this amoeba was isolated in *Gorilla gorilla gorilla*, *Pongo pygmaeus*), and old world primates, including a colobus monkey (*Colobus guereza kikuyuensis*) and a gibbon (*Hylobates concolor leucogenys*) [46,47,48], but have been reported in both immunocompetent and immunocompromised individuals of all ages. 

In this study, amoebae were isolated and identified at the general level based on morphologic features and at the species level by DNA sequencing. The diversity of FLA described agrees with previous studies [14,15,49,50]. Moura et al. reported the presence of *Acanthamoeba*, *Vahlkampfia*, *Hartmannella* spp. (currently named as *Vermamoeba* spp.), and *Echinamoeba* in a human fecal sample by culture-dependent methods. The authors demonstrated in vivo that the isolates of *Acanthamoeba* spp. were capable of causing cerebral lesions [14]. Indeed, an interesting observation was the detection of *Vannella* spp., which is not unprecedented. Even nonpathogenic *Vannella* sp. can be of clinical relevance, as they can act as vehicles for pathogenic organisms [31,32,33]. In support of this, *Vannella* stains and their endosymbionts (resembling microsporidia organisms and *Pseudomonas aeruginosa*) were detected from corneal scrapings [51]. In this study, the primers successfully amplified DNA from fan-shaped amoeba samples of the family Vannellidae, but no PCR products were successfully sequenced.

On the other hand, our findings of sequence analysis of the PCR products revealed the presence of the *Acanthamoeba* T4 genotype and *V. vermiformis *with high similarity to previous studies [52,53]. In general, morphological analyses pointed out mixed infections, which were corroborated with chromatogram sequence analyses. *Acanthamoeba* spp. have been classified into 23 genotypes (T1–T23) based on their 18S rRNA whole gene sequences. However, T4 genotype is the most prevalent in clinical and environmental samples [54]. In the present study, sequences were identical or similar to previously described isolates of the T4 genotype, a trend observed in previous studies. However, since the different genotypes of *Acanthamoeba* differ in their pathogenic potential, it is relevant to assess whether such differences exist among the subtypes/species within the same genotype, mainly the T4 genotype, which is related to the majority of *Acanthamoeba* infections. Indeed, the relatively low level of knowledge of global amoeba biodiversity indicates a very high probability of finding the species in any habitat. Several studies demonstrated the occurrence of potentially pathogenic *Acanthamoeba* and *Vermamoeba* in the oral and nasal mucosa of patients with suppressed immune status, such as patients with HIV/AIDS, patients undergoing hemodialysis, and healthy subjects [28,55,56].

In the last years, some studies have reported that the gut microbiome composition is shaped predominantly by environmental factors [57]. It seems reasonable to speculate that wild-caught and outdoor housed animals have the potential to be infected with a variety of bacterial organisms, protozoan, and metazoan parasites. Therefore, it is not unlikely that the diversity of FLA in feces may reflect the ubiquitous distribution of FLA in the environment (mainly soil and water collections). According to a study, a ubiquitous colonization and opportunistic infection by free-living eukaryotes such as Cercozoa, *Acanthamoeba* spp., and other Discosea were observed in pigs [16]. Strikingly, the authors reported that FLAs were recovered from stomach, ileum, caecum, colon, and rectum, and at both incubation temperatures of 25 °C and 37 °C. Diversity was dominated by amoebae: vahlkampfiids, vannellids, *Acanthamoeba* spp., *Hyperamoeba* sp., and *Vermamoeba vermiformis*.

It is worth noting that FLAs feed mainly on bacteria and are resilient to harsh abiotic factors. *Macaca mulatta* gastric pH, and acid output resemble that of humans [58]. So, how do FLAs survive through the stomach? Indeed, amoebae possess protection mechanisms against harsh environments. The capacity of the FLAs to adapt in any imaginable set of conditions is remarkable. However, this question is poorly understood and neglected.

In this study, we explored the impact of acid pH on the viability of *Acanthamoeba* spp. parasite growth rates, measured in range from pH 2 to 6. Surprisingly, at an acidic pH of 2, *A. castellanii* and *A. polyphaga* were observed to be moving for 24 h, but became nonviable within 48 h. In contrast, in pH ranging from 3 to 6, the amoebae remained viable for more than 96 h. A similar result was previously recorded with *Naegleria fowleri* [59]. Under harsh conditions, many amoebas survive by encystment. *Acanthamoeba* cysts have been shown to be resistant to extreme conditions including freezing, pH 2.0, and ultraviolet irradiations, heavy metal concentrations, desiccation, and when stored at a temperature of 4 °C over a long period of time (20 years) [60]. Likewise, many bacteria that transit the gastrointestinal tract (GIT) are known to be neutralophilic but can resist acidic pH using other approaches. For example, the ability of *H. pylori* to survive the low pH of the stomach would seem to suggest that it is an extreme acidophile. In fact, *H. pylori* is a neutrophile. This bacterium escapes acidity in the stomach by breaking down urea with enzyme-urease producing large amount of ammonia (alkaline) that raises the pH of the immediate environment, so that the pH increases to nearly neutral [61]. On the other hand, a study under in vitro conditions reported no effect on the viability of *A. castellanii* when cells were inoculated for encystation at different pH with or without the presence of 10% glucose and 50 mM MgCl_2_. In addition, an acidic pH (3.0) did not serve as a strong stimulus for the encystation of *A. castellanii*. In contrast, a neutral pH (7.0) was an optimum medium for approximately 30–40% encystation. Lastly, the authors reported that light-dark cycles, 5% CO_2_, and microaerophilic conditions had no effect on encystation of *A. castellanii* [62]. In the light of these observations, it is worth noting that FLAs feed mainly on bacteria and are resilient to harsh abiotic factors, which provide clear evidence that some FLAs might remain viable, consequently allowing colonization and persistence in gut tracts.

Although our findings revealed a diverse occurrence of FLAs infections, environmental contamination related to the samples collected can be excluded because part of the samples were collected postmortem from *Callithrix* spp. (wild PNH) and the washed gut of captive PNHs. Moreover, we analyzed serial samples on multiple non-consecutive days. Analyzing serial samples collected over several days can help distinguish between pseudo and natural parasites and contamination by soil. In our study, captive NPHs and free primates from the Atlantic Forest region are exposed to soil, water, and air and hence are at increased risk of infection by FLA. The risk of FLA acquisition is linked to the potential of the host being exposed to local environmental conditions. In many circumstances, NHPs under significant physiological stress or with a compromised immune system are at high risk of severe consequences and death from infectious disease, even those with low virulence. Infectious disease plays a major role in the lives of non-human primates and can have a tremendous negative impact. However, clinically relevant free-living amoebic diseases in animals are rare. Still, it can show severe and fatal progression, mainly due to a lack of awareness, leading to delayed diagnosis and a shortage of effective treatment.

There is a wide gap in our knowledge regarding FLAs in the mammalian gut, and questions arise as to whether they could represent long-term colonists or transient invaders in the mammalian intestinal microbiome. However, *Acanthamoeba* can also be found in the throat, intestine, and nasal region of healthy individuals [14,42,63]. Indeed, most healthy individuals (humans or other animals) do not develop disease despite regular contact with free-living amoebae. Presumably, the presence of antibodies to FLA in healthy human sera contributes to protection against infections. Asymptomatic exposures are common in healthy individuals [42,64,65]. In animal models, some studies in vivo and in vitro have shown effective protection against meningoencephalitis and keratitis after immunization with *Acanthamoeba* antigen, and anti-*Acanthamoeba* tear IgA provides an immunological barrier, blocking their adherence to epithelial cells [65]. Oral immunization protects against *Acanthamoeba* keratitis in corneal infections in pigs when administered before the corneal challenge [66] and in a hamster animal model [67].

In this study, most of the genera found have already been implicated in human and/or animal infections. However, our results showed only widespread asymptomatic exposures among non-human primates. Kollars and Wilhelm reported that wild mammals could become infected in nature and produce antibodies against *Naegleria* spp. in the manners described for laboratory experiments [68]. In general, exposure to antigens via mucosal surfaces in the gut induces the preferential generation of secretory IgA antibodies [69]. Although these are not the objectives of this study, we hypothesized that the interaction of FLA with mucosal surfaces (gut colonization) may induce a local and systemic humoral immune response that provides solid protection against developing intestinal and extraintestinal diseases in these animals.

Our findings do not allow for inferences on whether the detected FLA are residents or merely transients, nor do they indicate whether animals acquire natural immune responses. Beyond that, we did not characterize the pathogenicity of the isolates; we only detected a genotype considered potentially pathogenic, the *Acanthamoeba* T4 genotype. It is necessary to investigate the pathogenicity of the isolated strain. Consequently, future investigations have tremendous potential for understanding FLA in the gut, permitting further scrutiny of the natural immune responses of PNHs, chemotherapy, pathogenicity, and dynamics of parasite–host relationships.

## 5. Conclusions

A great diversity of FLAs was identified in the NHP microbiomes. To our knowledge, this is the first report presenting the occurrence of the potentially pathogenic *Acanthamoeba* T4 genotype and *V. vermiformis*, which represents a risk for primates. In addition, fan-shaped amoebae of the family Vannellidae and heterolobosean amoeboflagellates need additional workup and phylogenetic analysis to understand these amoebae’s genotypic diversity better. Crucially, further studies must focus on the molecular identification of isolates at the species level and evaluate their pathogenic potential.

## Figures and Tables

**Figure 1 tropicalmed-09-00108-f001:**
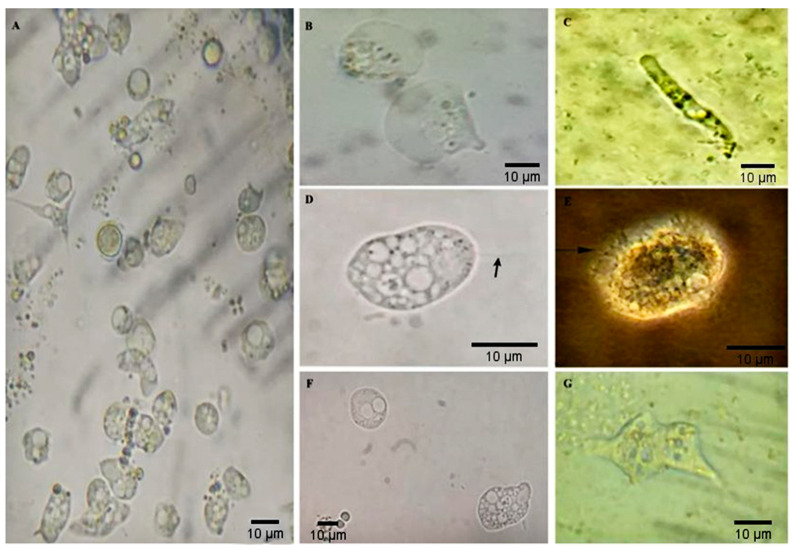
Microphotograph of unstained free-living amoeba trophozoites observed in cultures using a non-nutrient agar under light microscopy and phase contrast microscopy, bar = 10 μm. (**A**) Mixed-species infections where morphotypes of different amoebae were isolated on a non-nutrient agar plate, (**B**) Fan-shaped amoebae of the *family Vannellidae*; (**C**) An elongated cylindrical trophozoite of *Vermamoeba*-like amoeba; (**D**) heterolobosean amoebaflagellate, flagellate stage temporary pear-shaped with a pair of flagella (black arrows); (**E**) *Acanthamoeba* spp. show spine-like projections from the pseudopodia (acanthopodia); (**F**,**G**) unidentified trophozoite forms.

**Figure 2 tropicalmed-09-00108-f002:**
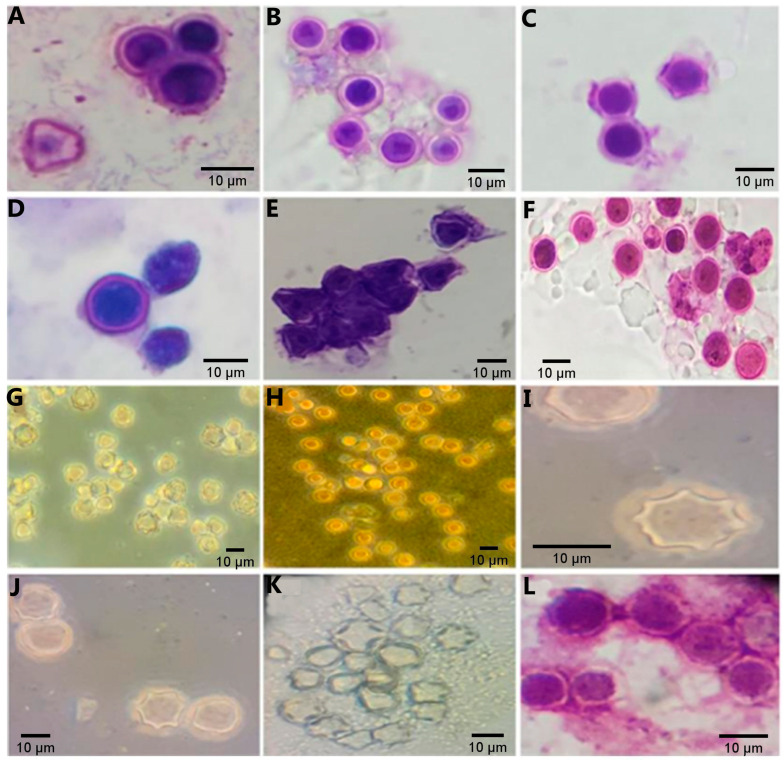
Microphotograph of free-living amoeba cysts found in fecal samples from non-human primates under light and phase contrast microscopy, bar = 10 μm. (**A**–**F**) rounded and polygonal cysts stained by Panoptic^®^ stain; (**G**,**H**) unstained cystic forms (round, triangle, and square) evidenced by *differential* interference contrast (DIC), suggestive of the genus *Acanthamoeba*; C-ectocyst (EC) and endocyst (ED); (**I**–**K**) = polygonal and stellate cysts, typical of the genus *Acanthamoeba* shows ectocyst (EC) and endocyst (ED). and (**L**) = unidentified amoeba with rounded cysts stained by Panoptic^®^ stain.

**Figure 3 tropicalmed-09-00108-f003:**
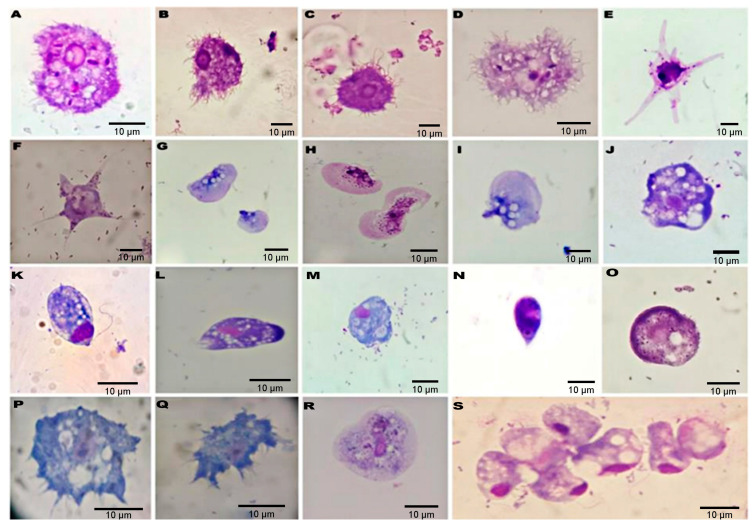
Microphotograph of free-living amoeba trophozoites found in fecal samples from non-human primates, stained by Panoptic stain, bar = 10 μm. (**A**–**D**,**O**–**Q**): *Acanthamoeba*-like trophozoites exhibiting fine short acanthopodia; (**E**,**F**): Flotation form of the *family Vannellidae*; (**G**–**I**): Fan-shaped amoebae of the *family Vannellidae;* (**K**,**N**): heterolobosean amoeboflagellates trophozoites; (**L**,**M**): The trophozoites from the heterolobosean amoeba assumed a monopodial form; (**J**,**O**,**R**,**S**): Unidentified trophozoites forms, showing a nucleus and several vacuoles.

## Data Availability

The datasets supporting the findings of this article are included within the article and its Appendix A. Sequences have been deposited in the GenBank database under accession numbers: OR68553, OR685532, OR685533, OR685534, OR685535, and OR685536.

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
