# Peer review of "Occurrence of Free-Living Amoebae in Non-Human Primate Gut"

_tropicalmed, 2024, doi:10.3390/tropicalmed9050108_

Round 1

Reviewer 1 Report

Comments and Suggestions for Authors

The article "Occurrence of free-living amoebae in non-human primate gut." submitted by Rodrigues Cardoso et al., is a very straightforward presentation of information. The information discussed in the article is important. there are some suggestions

1. Each Micrograph should have a scale bar. Magnification is confusing as pictures are cropped and adjusted digitally.

2. Discussion should be devoid of redundancy and may be more compact. 

3. It is not clear why only one PCR-amplified product was sequenced. (Page 8, line 249). Though it is written that only one is successfully sequenced it is unlikely that out of 33 amplicons, why only one? 

Comments on the Quality of English Language

There are a few typos. Particularly, some scientific names are not in italics either partially or fully. 

Author Response

Review 1

  1. Summary

We are grateful to the reviewer for taking time to carefully review the manuscript and give detailed and constructive comments, which has greatly helped to improve this paper. Below is our point-by-point response to each respective comment

  1. Questions for General Evaluation

The article "Occurrence of free-living amoebae in non-human primate gut." submitted by Rodrigues Cardoso et al., is a very straightforward presentation of information. The information discussed in the article is important. there are some suggestions

Comments 1: Discussion should be devoid of redundancy and may be more compact. 

Response 1 : We agree that the manuscript will benefit from a concise discussion. The discussion was rewritten and presents findings in the context of relevant literature with alternative interpretations considered as needed.

Point-by-point response to Comments and Suggestions for Authors

Comments 1: Each Micrograph should have a scale bar. Magnification is confusing as pictures are cropped and adjusted digitally.

Response 1 : I didn’t see it that way. Thanks for pointing out this inconsistency.   I have updated all the scale bars of the micrographs (Figure 1, Figure 2, and Figure 3).

Comments 2: It is not clear why only one PCR-amplified product was sequenced. (Page 8, line 249). Though it is written that only one is successfully sequenced it is unlikely that out of 33 amplicons, why only one? 

Response 2: All samples had mix-species infections where two or more morphotypes were in the same culture. The cloning methodology was applied, where only a single cell was seeded in a new culture medium. After that, DNA extraction and PCR reactions were performed in the culture pellet. However, analysis of the obtained sequencing electropherograms revealed overlapping peaks corresponding to different nucleotides, thus indicating co-infection (fan-shaped amoebae of the family Vannellidae).  Similarly, mixed infections were observed with DNA sequencing of Vermamoeba spp, but only one sample was identified as V. vermiformis. In this case, we may speculate on a preferential amplification of one target sequence over another (due to a low amount of DNA). Only the predominant DNA sequence was amplified and sequenced successfully, or we got to seed a single cell using the cloning methodology and succeeded with the amplification step and DNA sequencing. Moreover, the results are highly dependent on the quality of the primers used for amplification and it can not excluded that the effects of primer mismatches on amplification because  affect sequencing DNA producing  noise and double peaks.

Reviewer 2 Report

Comments and Suggestions for Authors

I believe that some improvement in English will make the meaning of the ms much easier to follow.

The introduction is helpful in understanding the field, but the discussion seems to be much longer than needed, perhaps twice as long. I think that increasing to focus to the results presented will help.

Comments on the Quality of English Language

While in general the meaning of the text can be understood, the English is choppy enough to be distracting, and in some cases the meaning/conclusions are hard to discern. Part of this seems to be due to English as a second language, and some as a simple lack of copy editing. While many small changes are needed, I believe this can all be easily addressed.

Author Response

Review 2

  1. Summary

We are grateful to the reviewer for taking time to carefully review the manuscript and give detailed and constructive comments, which has greatly helped to improve this paper.

  1. Questions for General Evaluation

Comments 1: I believe that some improvement in English will make the meaning of the ms much easier to follow.

Response 1: The manuscript was checked for the accuracy and consistency of grammar, spelling, tone, punctuation, terminology, and syntax. After editing the document, english became accurate and consistent, with expression clear and the overall readability of writing enhanced.

Comments 2: The introduction is helpful in understanding the field, but the discussion seems to be much longer than needed, perhaps twice as long. I think that increasing to focus to the results presented will help.

Response 2: We agree that the manuscript will benefit from a concise discussion. The discussion was rewritten and presents findings in the context of relevant literature with alternative interpretations considered as needed.

Comments 3: Comments on the Quality of English Language

While in general the meaning of the text can be understood, the English is choppy enough to be distracting, and in some cases the meaning/conclusions are hard to discern. Part of this seems to be due to English as a second language, and some as a simple lack of copy editing. While many small changes are needed, I believe this can all be easily addressed.

Response 3: The manuscript was checked for the accuracy and consistency of grammar, spelling, tone, punctuation, terminology, and syntax. The manuscript was  reviewed by a proofreading service

We would like to thank the referee again for taking the time to review our manuscript.

Reviewer 3 Report

Comments and Suggestions for Authors

This manuscript reports the presence of free living amoeba in the guts of non-human primates. Several previous studies have report the presence of these amoeba in the guts of humans and other animals, but this study has focussed on primates in a more or less natural state. Throughout the manuscript the term Naegleria–like is used but there is no sequence available for this strain and there is insufficient information to suggest that this strain is more like Naegleria than it is to any other Heterolobosean amoeboflagellate amoeba such as Vahlkampfia, Tetramitus or Psalteriomonas. Figures 1 and 2 state that the magnification is x400 but this is not helpful as the size of the image as viewed will differ.  It would be much more useful to include a 10um scale bar.  There is much inconsistency throughout in the use of italics for species names. These instances (not all of them) and other small problems are listed below.

Line 11. “health” not “healthy”.

Line 14. The term non-human primates should be defined where the abbreviation first appears not on line 17.

Line 16. “101” not “01”

Line 44. A comma after “Recently” not a full stop.

Line 48. It should be stressed here that the gut parasite Entamoeba does have a profound effect on humans ( Entamoeba histolytica kills >55,000 people each year) and other animals but despite it belonging to the Amoebozoa (like Vermamoeba and Acanthamoeba) it is not a free living amoeba (FLA) but is rather an obligate parasite.

Line 60. “not uncommon” is a very vague quantification and parasitologists generally consider AK as being an uncommon disease. Perhaps it would be safer to use the phrase “has been reported” instead of “being not uncommon” here.

Line 66. The sentence contains many errors. The word “idiopathic” is incorrect here. Perhaps the sentence would be better as “The symptoms of PAM and GAE can be mistaken for bacterial or viral diseases, and the clinicians who treat the patients may not familiar with diseases caused by FLA [29].”

Line 81. Italicise “Helicobacter pylori”.

Line 90. This sentence is incorrect and I suggest “The recent development of “culture-independent” methods of microbial characterization and high-throughput sequencing in fecal samples, have been very useful, but they do not directly answer the most critical questions of host–microbiota interactions.”

Line 108. “Euthanasia” does not require a capital E.

Line 111. The primate species “Macaca mulata, Saimiri ustus and Saimiri sciureus” should be italicised. Also it may be helpful to give the common name for these species (Rhesus macaque, Bare-eared squirrel monkey and the South American Squirrel monkey).

Line 116. Italicise “Callithris”.

Line 125. Italicise “Escherichia coli”  here and line 136

Line 149. Italicise “Acanthamoeba” and check other binomials throughout the manuscript.

Line 175. The use of the phrase “potentially pathogenic” is a little misleading here as no tests have been conducted on the properties of these FLA, pathogenic or otherwise. Genus Vannella contains no known pathogens.

Line 182. “even if anti-fungal drugs were used.” Instead of “even if when added anti-fungal drugs.”

Line 185. This sentence is about the Naegleria-like FLA but this is not made clear. The sentemce would be clearer as “The trophozoites of the Naegleria-like FLA assumed a monopodial form, and the amoeboid-form organism changed to the transient, flagellate form with flagella at the broad end (Fig. 1d)”.

Line 188. The cyst of Naegleria is a single walled structure, not a double walled structure. However, some Naegleria species have an outer gelatinous layer.

Line 196 “pine-like pseudopods” is incorrect. Acanthopdia are projections that arise from the pseudopod and they are spine shaped not pine shaped, so the sentence should be “E: Acanthamaoeba spp. show spine-like projections from the pseudopodia (acanthopodia),…”

Line 206 “contrast” not “contrst”.

Line 208 “Differential” should not be in italics.

Line 210 “shows” should not be in italics.

Line 223. The different stages of Heterolobose organisms are cysts, trophozoites or flagellates. The phrase should be “The trophozoites from the Heterolobosean amoeba assumed a monopodial forms”

Line 232. The word family should not be italicised. Also in line 305.

#Line 237. What are AVLs her? Is this FLAs? Also what is PAGE on the next line?

Line 253. “amoeboflagellates” instead of “flagella”.

Line 311. “Vannella” not “Vanella”.

Line 391. The word “Notoriously” is wrong here (and elsewhere).

Comments on the Quality of English Language

None

Author Response

Review 3

  1. Summary

We are grateful to the reviewer for taking time to carefully review the manuscript and give detailed and constructive comments, which has greatly helped to improve this paper.

  1. Questions for General Evaluation

Comments 1: This manuscript reports the presence of free living amoeba in the guts of non-human primates. Several previous studies have report the presence of these amoeba in the guts of humans and other animals, but this study has focused on primates in a more or less natural state. Throughout the manuscript the term Naegleria–like is used but there is no sequence available for this strain and there is insufficient information to suggest that this strain is more like Naegleria than it is to any other Heterolobosean amoeboflagellate amoeba such as Vahlkampfia, Tetramitus or Psalteriomonas. Figures 1 and 2 state that the magnification is x400 but this is not helpful as the size of the image as viewed will differ.  It would be much more useful to include a 10um scale bar.  

There is much inconsistency throughout in the use of italics for species names. These instances (not all of them) and other small problems are listed below.

Response 1 : We agree with your commentary.  We have fixed the errors.

Thanks for pointing out this inconsistency. I have updated all the scale bars of the micrographs (figure 1, figure 2 and figure 3).

The manuscript was checked for the accuracy and consistency of grammar, spelling, tone, punctuation, terminology, and syntax. The manuscript was reviewed by a proofreading service

We fear the reviewer may have misunderstood us here. In a general outline, the diversity of free-living amoebae remains under-explored. Classical identification of heterolobosean species relied on morphological characters. However, morphology presented insufficient features for reliable species or genera differentiation and recognition. Typical heteroloboseans are amoeboflagellates that have distinct amoeba and flagellate stages. In most species, the amoeba stage can also transform into a cyst, resulting in a characteristic three-phase life cycle. Flagellates usually possess two or four flagelles, with amoebae forming the central phase. Psalteriomonas present at the apical side of the cell two of the four flagella clusters; Tetramitus present flagellates that usually have a distinctive conical shape with four equal-length flagella that beat heterodynamically, and Naegleria spp present flagellated phase with biflagellate similar our findings (see additional file 1). However, we agree with the reviewer that we can not state that it is the genus Naegleria because the DNA sequence was not obtained, so named as Naegleria-like.

  1. Point-by-point response to Comments and Suggestions for Authors

Comments 1: Line 11. “health” not “healthy”.

Response 1: Thanks, We have fixed the error

Comments 2: Line 14. The term non-human primates should be defined where the abbreviation first appears not on line 17.

Response 2:Thanks, We have fixed the error

Comments 3: Line 16. “101” no t “01”

Response 3: Thanks, We have fixed the error

Comments 4: Line 44. A comma after “Recently” not a full stop.

Response 4:Thanks, We have fixed the error

Comments 5: Line 48. It should be stressed here that the gut parasite Entamoeba does have a profound effect on humans (Entamoeba histolytica kills >55,000 people each year) and other animals but despite it belonging to the Amoebozoa (like Vermamoeba and Acanthamoeba) it is not a free living amoeba (FLA) but is rather an obligate parasite.

Response 5: Amoebozoa is divided into two major subclasses: Class Lobosea and Class Conosa.  Pathogenic genera within Class Lobosea are capable of producing meningoencephalitis and class Conosa includes one pathogenic specie  Entamoeba histolytica, which typically produces a dysentery-like condition.

Comments  6: Line 60. “not uncommon” is a very vague quantification and parasitologists generally consider AK as being an uncommon disease. Perhaps it would be safer to use the phrase “has been reported” instead of “being not uncommon” here.

Response 6: We agree with this and have incorporated your suggestion throughout the manuscript. We rewrote the sentence, “In recent years, the frequency of Acanthamoeba keratitis has increased worldwide after extended wear of soft contact lenses.

Comments 7:Line 66. The sentence contains many errors. The word “idiopathic” is incorrect here. Perhaps the sentence would be better as “The symptoms of PAM and GAE can be mistaken for bacterial or viral diseases, and the clinicians who treat the patients may not familiar with diseases caused by FLA [29].”

Response  7: Thanks, We agree with this. We rewrote the sentence,  “The symptoms are non-specific which can be mistaken for other bacterial and viral diseases, and the clinicians who treat the patients are not familiar with FLA”

Comments  8:Line 81. Italicise “Helicobacter pylori”.

Response 8: We agree with this and have incorporated your suggestion in the manuscript.

Comments  9: Line 90. This sentence is incorrect and I suggest “The recent development of “culture-independent” methods of microbial characterization and high-throughput sequencing in fecal samples, have been very useful, but they do not directly answer the most critical questions of host–microbiota interactions.”

Response 9: We agree with this and have incorporated your suggestion in the manuscript.

Comments  10:Line 108. “Euthanasia” does not require a capital E.

Response  10: Thanks, We have fixed the error

Comments 11: Line 111. The primate species “Macaca mulata, Saimiri ustus and Saimiri sciureus” should be italicised. Also it may be helpful to give the common name for these species (Rhesus macaque, Bare-eared squirrel monkey and the South American Squirrel monkey).

Response 11: Thanks, we have incorporated your suggestion in the manuscript.

Comments 12: Line 116. Italicise “Callithris”.

Response  12: Thanks, we have incorporated your suggestion in the manuscript.

Comments  13: Line 125. Italicise “Escherichia coli”  here and line 136

Response 13: Thanks, We have fixed the error

Comments  14:Line 149. Italicise “Acanthamoeba” and check other binomials throughout the manuscript.

Response 14:Thanks, We have fixed the error

Comments  15: Line 175. The use of the phrase “potentially pathogenic” is a little misleading here as no tests have been conducted on the properties of these FLA, pathogenic or otherwise. Genus Vannella contains no known pathogens.

Response 15: We agree with reviewer.  we have incorporated your suggestion in the manuscript.

Comments  16: Line 182. “even if anti-fungal drugs were used.” Instead of “even if when added anti-fungal drugs.”

Response  16: Thanks, we have incorporated your suggestion in the manuscript.

Comments  17:Line 185. This sentence is about the Naegleria-like FLA but this is not made clear. The sentemce would be clearer as “The trophozoites of the Naegleria-like FLA assumed a monopodial form, and the amoeboid-form organism changed to the transient, flagellate form with flagella at the broad end (Fig. 1d)”.

Response  17:Thanks, we have incorporated your suggestion in the manuscript.

Comments  18: Line 188. The cyst of Naegleria is a single walled structure, not a double walled structure. However, some Naegleria species have an outer gelatinous layer.

Response  18: Thanks, we have incorporated your suggestion in the manuscript.

Comments  19: Line 196 “pine-like pseudopods” is incorrect. Acanthopdia are projections that arise from the pseudopod and they are spine shaped not pine shaped, so the sentence should be “E: Acanthamaoeba spp. show spine-like projections from the pseudopodia (acanthopodia),…”

Response  19: Thanks. Typographical errors were corrected.

Comments  20: Line 206 “contrast” not “contrst”.

Response  20: Thanks. Typographical errors were corrected.

Comments  21:Line 208 “Differential” should not be in italics.

Response  21: Thanks. This typographical error was corrected.

Comments  22:Line 210 “shows” should not be in italics.

Response  22: Thanks. This typographical error was corrected.

Comments  23: Line 223. The different stages of Heterolobose organisms are cysts, trophozoites or flagellates. The phrase should be “The trophozoites from the Heterolobosean amoeba assumed a monopodial forms”

Response  23: Thanks, we have incorporated your suggestion in the manuscript.

Comments  24: Line 232. The word family should not be italicised. Also in line 305.

Response  24: Thanks. This typographical error was corrected.

Comments 25: Line 237. What are AVLs her? Is this FLAs? Also what is PAGE on the next line?

Response  25: Thanks, we corrected these mistakes.

Comments  26: Line 253. “amoeboflagellates” instead of “flagella”.

Response  26: Thanks, we have incorporated your suggestion in the manuscript.

Comments  27: Line 311. “Vannella” not “Vanella”.

Response  27: Thanks, we corrected this  mistake.

Comments 28 : Line 391. The word “Notoriously” is wrong here (and elsewhere).

Response  28: Thanks, we have incorporated your suggestion in the manuscript and the sentence were modified.

Round 2

Reviewer 2 Report

Comments and Suggestions for Authors

I'm not sure that the English has been much improved since the last draft. Below are examples (only examples) of problematic passages.

"Indeed, an interesting observation was the detection of Vannella spp, which is not one fact unprecedented."

"In turn, when used the set of primers...:"

"...fungal heavily over-growth.."

"Flotation forms of amoebae (Figure 3 E-F) and fan-shaped amoebae of the family Vannellidae..."

"All sample positives were mixed with their different genera."

"...FLAs were recovered in from stomach..."

"...have already been involved in humans and, or animals infectious."

Comments on the Quality of English Language

It's not clear to me that any attempt was made to improve the readability of the MS. Perhaps they can try again?

Author Response

Dear

The Manuscript has undergone thorough proofreading and editing by Edit Syndicate. It is deemed to be free of grammatical errors and suitable for publication as a standard scientific report
